# Management of Osteoblastoma and Giant Osteoid Osteoma with Percutaneous Thermoablation Techniques

**DOI:** 10.3390/jcm10245717

**Published:** 2021-12-07

**Authors:** Antonio Izzo, Luigi Zugaro, Eva Fascetti, Federico Bruno, Carmine Zoccali, Francesco Arrigoni

**Affiliations:** 1Department of Biotechnological and Applied Clinical Sciences, University of L’Aquila, 67100 L’Aquila, Italy; e.fascetti@inail.it (E.F.); federico.bruno.1988@gmail.com (F.B.); 2Department of Emergency and Interventional Radiology, San Salvatore Hospital, 67100 L’Aquila, Italy; luigi.zugaro@virgilio.it (L.Z.); arrigoni.francesco@gmail.com (F.A.); 3Orthopaedics and Traumatology Unit, Policlinico Umberto I, Sapienza University of Rome, 00185 Roma, Italy; carminezoccali@libero.it

**Keywords:** Osteoblastoma (OB), thermoablation techniques, musculo-skeletal interventional radiology

## Abstract

Osteoblastoma (OB) is a rare, benign bone tumor, accounting for 1% of all primary bone tumors, which occurs usually in childhood and adolescence. OB is histologically and clinically similar to osteoid osteoma (OO), but it differs in size. It is biologically more aggressive and can infiltrate extraskeletal tissues. Therapy is required because of severe bone pain worsening at night. Moreover, non-steroid anti-inflammatory drugs (NSAIDs) are not a reasonable long-term treatment option in young patients. Surgical excision, considered the gold standard in the past, is no longer attractive today due to its invasiveness and the difficulty in performing a complete resection. The treatment of choice is currently represented by percutaneous thermoablation techniques. Among these, Radiofrequency ablation (RFA) is considered the gold standard treatment, even when the lesions are located in the spine. RFA is a widely available technique that has shown high efficacy and low complication rates in many studies. Other percutaneous thermoablation techniques have been used for the treatment of OB, including Cryoablation (CA) and laser-ablation (LA) with high success rates and low complications. Nevertheless, their role is limited, and further studies are necessary.

## 1. Introduction: Clinical, Pathological, and Imaging Findings

Osteoblastoma (OB) is a rare, benign bone tumor, accounting for 1% of all primary bone tumors and 3% of all benign bone tumors [1,2,3]. They usually occur in childhood and adolescence.

Jaffe and Mayer are credited with identifying it as an entity in 1932; subsequently, OBs have been described as “giant osteoid osteomas,” emphasizing their close histologic resemblance to osteoid osteoma (OO), as well as their larger size.

OBs are histologically similar to OOs; for this reason, some authors subscribe to the concept that they represent different clinical expressions of the same pathological process [1,4]. However, they differ in clinical presentation, radiological features and natural histories. In fact, whereas OOs tend to regression, OBs tend to get worse. Possible, although rare, malignant transformations have in fact been reported [1,2].

Generally, OBs are larger in size than OOs. Lesions presenting with a nidus greater than 1.5 cm of diameter are conventionally considered osteoblastomas. They present less sclerotic borders and a thin periosteal reaction surrounding the lesion [4,5].

Similarly to OO, OB is most often found in patients in their first to third decades, more commonly found in males, with a male to female ratio of 2:1. OB is four times less frequent than OO, and, even if the long tubular bones are frequently affected, the lesion has a preference for the posterior structures of the vertebral column. Among the long tubular bones, the most common bones involved are the femur and tibia [1,3,6].

The clinical presentations of OB and OO are different. Some patients are asymptomatic, and the lesion is found incidentally. Symptomatic patients generally present with localized pain, usually less severe than OO; pain rarely interferes with sleep and does not appear to be as readily relieved by salicylates as in osteoid osteoma [1,2,6]. The main clinical manifestations of the lesions involving the posterior aspect of the spine are of neurological nature and include progressive focal or radicular pain exacerbated by movement; scoliosis is also frequently associated.

Conventional Radiography helps identify OBs in most cases. The interior aspect of the lesion appears radiolucent or with various degrees of internal ossification. Most tumors are spherical or slightly oval; the tumor margin is well defined and in half of the cases it is possible to identify considerable reactive sclerosis. In the great majority of tumors, the cortex is perfectly intact, although expanded and thinned (75%) [1,2,4,7]. Close to three out of five OBs show periosteal new bone formation, which in some cases is extremely prominent, and generally regular and of the solid type.

Radiographically, OBs may appear (a), almost identical to osteoid osteoma, but much larger; (b) as an expansive lesion similar to an aneurysmal bone cyst, with small radiopacities in the center; (c) as an aggressive lesion, simulating a malignant tumor.

Conventional tomography (CT) is the gold standard for the diagnosis and management of OB: in most cases, the nidus appears as an osteolytic area, or it presents with various degrees of internal calcifications in the remaining cases. In 50% of cases, it is possible to identify considerable reactive sclerosis, which may appear regular and thick. In the great majority of tumors, the cortex is perfectly intact, although expanded and thinned. Close to three out of five OBs show periosteal new bone formation. CT scans allow the evaluation of the real size of the lesion, the presence and extension of perilesional sclerosis, the presence and characteristics of periosteal reaction and relations with the surrounding anatomical structures (Figure 1a,b).

Magnetic Resonance (MR) has a limited role in the diagnosis of OBs. On the MR images, the signal intensity of OBs is similar to those of the majority of bone tumors: SE Tl-weighted images show a low-to-intermediate signal intensity, while on the T2- weighted images, the signal intensity is intermediate-to-high. MR can reveal peritumoral edema in the adjacent marrow and in soft tissues and may show associated disorders, such as inflammatory synovitis [1,3,8].

## 2. Treatment

OBs are benign bone lesions; nevertheless, they require therapy to alleviate severe bone pain as an alternative to NSAIDs which are not a reasonable long-term treatment option in young patients due to undesirable side effects. In the past, surgical excision was the preferred therapy [7,9,10,11,12], but currently, it is no longer attractive due to its invasiveness and difficulty in performing a complete resection without the risk of damaging the sensitive structures of the spine. For these reasons, today the treatments of choice of these lesions are represented by percutaneous thermoablation techniques, among which Radiofrequency ablation under CT-guidance (RFA) is the gold standard.

### 2.1. Literature Research

For our review, we searched MEDLINE, MEDLINE In-Process, EMBASE, and the Cochrane databases from 1992 to 2020, targeting our search based on condition and intervention using the keywords “osteoblastoma”, “osteoma”, “benign bone lesion”, “ablation”, “thermoablation” combined in appropriate algorithms. In addition, the biographies of the resulting articles were screened for further inclusion.

All studies matching the following criteria were eligible for inclusion: (1) prospective or retrospective cohort study for percutaneous treatment of OB or OO under CT guidance or MR guidance; (2) patient OB diagnosis by way of at least one CT or MR examination; (3) RFA, Cryoablation, MRgFUS and other thermoablation techniques as the object of the study; (4) English language; (5) follow-up ≥ 12 months.

### 2.2. Surgical Treatment

In the past, surgical resection or curettage was the only therapeutic alternatives to conservative treatment. Surgical resection has reported a success rate ranging from 88% to 97% [4,9,13,14], similar to that of thermoablation techniques. Usually, surgery is performed under general anesthesia. The accurate localization of the lesion is performed with fluoroscopy.

Surgery consists of intralesional excision (curettage) or total excision. In addition to thorough decompression of the spinal cord and nerve roots, total excision of the involved posterior elements is performed. When the vertebral body is involved, intralesional excision (curettage) is performed to ensure maximal removal of the tumor and to minimize the risk of tumor recurrence [4,13,15]. Reconstruction of the stability and structure of the spine is selectively performed based on the extent of the lamina and vertebral body resection required, namely, in presence of removal of the pedicle and/or injury of the facet. Though successful in the majority of cases, surgery has several drawbacks, including the inability to reduce the resection margins, the need for rehabilitation, the risk of pathological fractures, the need for general anesthesia, and the resulting decrease in patient satisfaction (50% of patients are dissatisfied) [7,14]. Moreover, surgery requires hospitalization, which increases health costs. All these reasons lead to the choice of thermoablation techniques to treat OBs.

### 2.3. Radiofrequency Ablation (RFA)

Currently, Radiofrequency ablation (RFA) under CT-guidance is accepted as the gold standard treatment for OB, even when presenting in the spine, where, if necessary, it is used with thermal protection techniques [3,7,8,9,10,14,15,16,17].

#### 2.3.1. Thermoablation: Physical Aspects

In RFA, a radiofrequency (RF) generator is used to deliver high-frequency, alternating current (375–600 kHz) to the patient through an RF probe. The current passes through the exposed active tip of the probe and results in the oscillation of charged tissue molecules (ions) within the ablation zone, producing frictional heat. The thermal effect depends on the electrical conducting properties of the treated tissue and the characteristics of the RF probe. When a local tissue temperature between 60 °C and 100 °C is reached, there is a protein denaturation and immediate coagulative necrosis. There are several RFA probes. The unipolar systems use dispersing grounding pads placed on the patient skin near the ablation site to serve as a receiving limb of the electrical circuit to prevent potential skin burns while bipolar systems use built-in transmitting and receiving electrical elements within the probe. The choice of the RF probe depends in large part on the volume of tissue to be ablated and to the proximity to critical structures. The main advantage of RFA is the precise determination of the morphology of the ablation zone beyond which tissues are safe from thermal injury. RFA is primarily used for the treatment of lesions that are mainly osteolytic because the higher intrinsic impedance of sclerotic bone lesions prevents the radiofrequency circuit from generating sufficiently high temperatures to ensure cell death and renders RFA ineffective [18,19,20]. The procedure is similar to that used for the treatment of OO, but the area of ablation is larger and it is necessary to be careful to avoid the sensitive structures of the spine. RFA uses the heat produced by radiofrequencies to induce controlled necrosis of the nidus.

#### 2.3.2. Thermoablation: Technical Aspects

The procedure is generally carried out under CT-guidance, which allows proper visualization of the lesion and guides the positioning of the electrodes. Different anesthetic approaches are used, depending on lesion location and patient age, but the most effective ones are spinal anesthesia and local plexus block [21].

The external access allows reaching the nidus through the shortest route possible, avoiding the critical anatomical structures. To reach the nidus, bone biopsy needles of different sizes are used, depending on the depth of the lesion and the size of the electrode used. After the biopsy is performed, the needle is removed and the electrode probe is coaxially introduced. The electrodes used have an exposed tip with adequate dimensions to cover the entire lesion. An adequate distance from vital structures should be ensured. The standard safety distance is at least 1 cm. The majority of operators use a device for temperature control. In this way, the temperature reaches 90 °C and is maintained for 4 or 6 min, till complete and radical ablation [17] (Figure 2a,b). After treatment, the patients can load weight on the treated area. Nocturnal pain symptoms recede immediately, but an infusion pump of analgesic drugs is often necessary during the first hours.

A successful OB treatment depends on the exact localization of the lesion because the temperature significantly decreases after more than 1 cm of distance from the active tip. Moreover, the cortical bone has an insulating effect [14,22]. When the cortical bone is not preserved and the distance to the neural structures is ≤10 mm, RFA may benefit from thermal protection techniques [11,14,23]. There are many thermoprotective measures, including carbodissection or hydrodissection, thermocouples, and nerve root electrostimulation [23]. Carbo- and hydrodissection allow the displacement and/or insulation of vulnerable structures by injecting CO2 or fluid through needles positioned between the planned thermoablation margin and the adjacent critical structures, creating the distance necessary for a safe ablation. Whenever fluid dissection is required in ablation performed by radiofrequency energy, dextrose in water (D/W 5%) is preferred over saline solution because of the high electrical conductivity of the latter [24].

Thermocouples are deployed around structures at risk to monitor local temperature changes; electrostimulation allows safe monitoring of motor nerve conductivity by positioning a stimulating electrode in contact with the nerve root proximal to the zone of ablation [24].

#### 2.3.3. Clinical Aspects

RFA shows high efficacy in the treatment of OB, with a reported primary technical success rate of 90–100% and primary clinical success rate of 89.5–95% [3,7,8,9,14,25,26], slightly lower than in the treatment of OO (primary technical and clinical success rates of 95–100% and 90–100%, respectively) [7,9,10,15,27]. The symptomatic recurrence after a period of comfort is not uncommon and is between 10–44% [7,9,14]. Reasons for unfavorable outcomes after RFA ablation are not yet fully understood. There is scarce literature identifying the risk factors for OB symptomatic recurrences that include the elongated morphology of the lesion, not allowing adequate coverage by the probes, the excessively irregular shape of perilesional sclerosis, and the young age of patients [28].

There are no significant differences in success rates between surgery (88–97%) and RFA, but the latter is a better tool in treating symptomatic recurrences after surgical resections (95–100% vs. 90%) [3,7,10,14]. Moreover, RFA has lower complication rates (0–6%), and, when present, complications are minor and resolve in a few days [7,10,25]. Another advantage of RFA over surgery is represented by the high indicators of patient satisfaction (90–92% of patients are satisfied vs. 50% of surgery). The days of hospitalization are fewer (median values 8 vs. 2 days, respectively) and the total procedural costs lower (7596 € vs. 3663 €, respectively) [7,14].

The complication rates are low (range 0–6%) [3,7,10,25]. The most commonly observed complications include skin burning in the treated site, followed by local infection, necrosis of the adjacent structures and post-intervention fractures. Minor complications are loss of sensitivity in the treated site that generally improves spontaneously in about 2 weeks. The complication rate is closely associated with the presence of critical anatomical structures near the lesion. The risks, however, are reduced by employing thermoprotective measures.

RFA is, therefore, considered the gold standard for the treatment of OBs, while surgery is currently the least useful for the purpose.

### 2.4. Cryoablation (CA)

Cryoablation (CA) belongs to the thermoablation techniques and may be employed in the treatment of Osteoblastomas (OB). Many authors have reported high efficacy and low complication rates [20,23,29,30]; however, the experience with this technique is still limited, despite its multiple theoretic advantages, including real-time monitoring of the ablation zone (i.e., ice ball), minimal post-procedural pain, and capability of the ice ball to transmit across sclerotic bone [31].

#### 2.4.1. Thermoablation: Physical Aspects

Cryoablation (CA) utilizes cold thermal energy (freezing) to achieve tumoral destruction: current systems use argon and helium gases delivered through small cryoprobes to induce rapid freezing and thawing of target tissues. Extreme hypothermia propagates through the surrounding tissues to form a localized ice-ball around the probe tip. The precise mechanisms of cryogenic cellular destruction are complex, but primarily involve direct physical damage and vascular-mediated cytotoxicity [20,32].

#### 2.4.2. Thermoablation: Technical Aspects

The CA procedures are performed under CT-guidance, with different anesthesiological approaches (loco-regional anesthesia, sedation-analgesia, general anesthesia). The number and type of cryoprobes are selected on the basis of preoperative CT assessment of OB size and morphology. When multiple probes are used, they are spaced 0.5–1 cm apart. Probes are deployed coaxially through bone trocars.

CA is performed using different freezing protocols according to the size of the target OB and the need for effective protection of nearby non-target structures. Freezing cycles are terminated prematurely or reduced in power if the ice ball extends within 1 cm of the adjacent critical structures and ancillary thermoprotective measures alone are deemed insufficient for a safe procedure. Intermittent CT images are obtained to monitor the ice ball extension during the freezing phases. When necessary, one or more ancillary thermoprotective measures, including carbodissection or hydrodissection (or both), thermocouples, and nerve root electrostimulation are used to protect the nearby (<1 cm) nontarget structures.

#### 2.4.3. Clinical Aspects

In different studies, CA has shown high efficacy in the treatment of OBs, with technical success (i.e., when the ice ball completely encompasses the lesion with an additional safety margin of 2–3 mm) of 100%, and primary clinical success (i.e., patients who reported complete pain relief) of 100% at 1 month and 78% at 12 months [20,23,28,29,31,33].

Complication rates are low and associated with the site of OB, in particular the closeness to neural structures. However, neural complications are rare and generally resolve in a few days with high doses of steroid therapy, thus suggesting the presence of an inflammatory component. Most of the complications are minor and include skin burning in the treated site, followed by local infection and loss of sensitivity in the treated area.

In conclusion, percutaneous Cryoablation is an effective and safe therapeutic option for patients presenting with painful OBs, in alternative to RFA, and with some advantages, which include the possibility to visualize in real-time the extension of the ice ball, thus providing the operators with an additional method to prevent injuries to the nearby non-target structures; the possibility to deploy more than one cryoprobes with a synergistic effect; to facilitate the formation of an ice-ball precisely adapted to the OB size and morphology; the capability to perform the procedure even in case of heavily mineralized OBs because the ice ball radiates with ease through the sclerotic bone, contrary to RFA, whose effect is hindered by high impedance of heavily sclerotic lesions. However, further studies are needed to validate the clinical use of this technique.

### 2.5. Other Thermoablation Techniques

Laser-ablation uses optical fibers to transmit infrared light energy into a tumor to produce rapid temperature elevation, protein denaturation, and coagulative necrosis. A continuous wave semiconductor diode laser delivers energy to the tumor by using a flexible single-use bare-tipped optical fiber with polymer cladding placed coaxially. The amount of energy to be delivered is calculated according to the formula [Nidus Size (mm) × 100 Joules + 200 Joules], and the duration of the ablation is typically 200–600 s, depending on the nidus size [31,34].

There are a few experiences in the treatment of OBs with Laser-ablation, however, some authors have used this technique for the treatment of Osteoid Osteomas with high efficacy and low complication rates [35,36]. Gangi et al. treated 114 OOs with laser-ablation, with pain relief in 112 out of 114 patients, six cases of pain recurrences and only one major complication (reflex sympathetic dystrophy). Therefore, with caution, this technique may be used, as an alternative therapeutic option, also for the treatment of OBs.

Microwave ablation uses antennae to deliver electromagnetic microwaves (approximately 900 MHz) to target tissue, which results in the agitation of ionic molecules and frictional heat and, subsequently, tissue coagulative necrosis. Although the hypoattenuating ablated tissue is often identified on CT, the margins of the ablation zone are not well-defined; this is considered a general disadvantage for the use of microwaves on spinal lesions and Osteoblastomas [20,31].

### 2.6. MRgFUS

#### 2.6.1. Thermoablation: Physical and Technical Aspects

This technique uses MR to localize the lesion. A specific cradle with a built-in ultrasound generator called a transducer replaces the standard MRI cradle during the treatment. The treatment consists of a series of energy deliveries (sonications) into the nidus. During each sonication, the specific MR sequences acquired provide real-time thermometric maps of the region of interest, to avoid thermal damage of the surrounding tissues. Each sonication is considered effective when a minimum temperature of 60 °C is reached in the nidus or in the closest soft tissues [5,8,37]. The treatment is considered complete when the entire lesion plus a 5-mm safety margin is covered with sonications (Figure 3).

#### 2.6.2. Clinical Aspects

MRgFUS has been described in different studies as bringing a clinical success range of 89–100% with no pain after 1-year follow-up, low complication rates, and no reported complications [5,38,39,40,41].

This technique presents multiple advantages. First, there is no need to use needles to perform incisions on the skin; second, there is no exposure to radiation; both factors are advantageous in the treatment of children and adolescents. Moreover, it is possible to adapt the ultrasound beam to the morphology and the size of OBs, which may be entirely covered by sonications.

The limits of the technique include the impossibility to perform a biopsy. It is worth considering that, though rare, the malignant transformation of OB in osteosarcoma is an event reported in the literature. For this reason, MRgFUS should be reserved for osteoblastomas that show clear radiological signs of benignity, including the presence of the nidus, and a not expanding nor growing appearance [42].

Other limitations include that only superficial lesions can be treated, it is necessary to have an adequate acoustic window, as well as the availability of a high magnetic field MRI unit.

In conclusion, when available, MRgFUS is a valid alternative therapeutic option in the treatment of osteoblastomas, especially in pediatric patients.

## 3. Conclusions

Osteoblastoma (OB) is a rare, benign bone tumor, accounting for 1% of all primary bone tumors and 3% of all benign bone tumors. Therapy is always required to alleviate severe bone pain worsening at night. Moreover, NSAIDs are not a reasonable long-term treatment option in young patients due to undesirable side effects.

In the past, surgical excision was the preferred therapy but at the time of writing it is no longer attractive due to the high rates of recurrences and complications, the need for rehabilitation and hospitalization, and the risks of pathological fractures.

For all these reasons, today, the treatment of choice of OBs are the percutaneous thermoablation techniques, and among these, Radiofrequency ablation (RFA). RFA is a widely employed technique, bringing high technical and clinical success rates and low complication rates. In addition, it has good indicators of patient satisfaction, requires a few days of hospitalizations which reduces the procedural costs. For all these reasons, it is considered the gold standard for the treatment of these lesions.

Cryoablation has shown high efficacy and low complication rates with some advantages, such as the possibility to visualize in real-time the extension of the ice ball, to deploy multiple cryoprobes creating a synergistic effect, and to perform the procedure even in the case of heavily mineralized OBs. Cryoablation is, therefore, a valid therapeutic option, but further studies are necessary. At the time of writing, there is scarce experience with other thermoablation techniques. MRgFUS has shown high efficacy and low complication rates. Nevertheless, it cannot be widely employed and is applied only on lesions located on the bone surface.

## Figures and Tables

**Figure 1 jcm-10-05717-f001:**
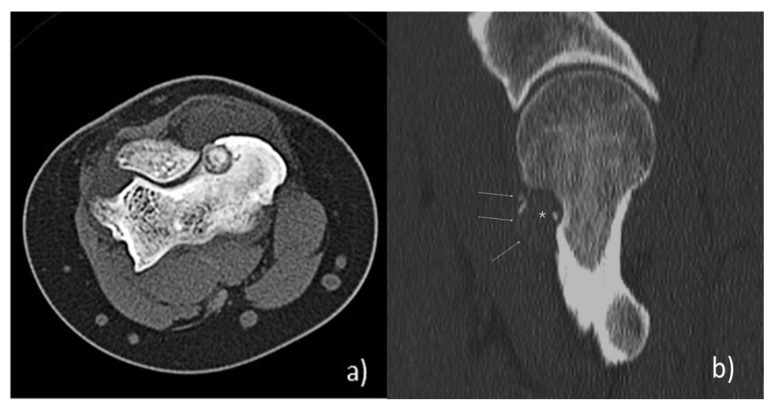
(**a**) Axial-CT images of intra-articular OB localized at the humeral epicondylus. The “nidus” has a rounded shape and well defined tumor margins, with high degree of internal calcifications; no presence of perilesional sclerosis. (**b**) Sagittal-CT images of OB localized at the femoral neck, with expansive aspect: the “nidus” is entirely radiolucent, with no sclerosis and small radio-opacities; the bone cortex is perfectly intact, although expanded and thinned.

**Figure 2 jcm-10-05717-f002:**
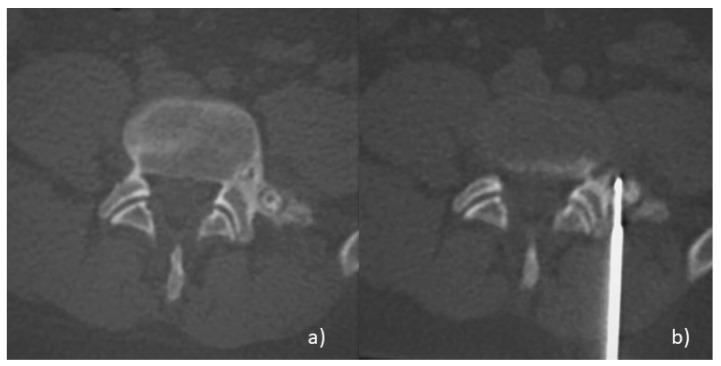
Radiofrequency-ablation (RFA) of a vertebral Osteoblastoma (OB). (**a**) Axial-CT image of OB located in the transverse process of a lumbar vertebra, near the spinal root. (**b**) Axial-CT image during the treatment: under CT-guidance, an RF probe with an exposed tip is perfectly located into the nidus; it is possible to achieve the complete and radical ablation of the lesion. A safety distance of about 1cm is ensured, between the exposed tip and the adjacent critical anatomical structures.

**Figure 3 jcm-10-05717-f003:**
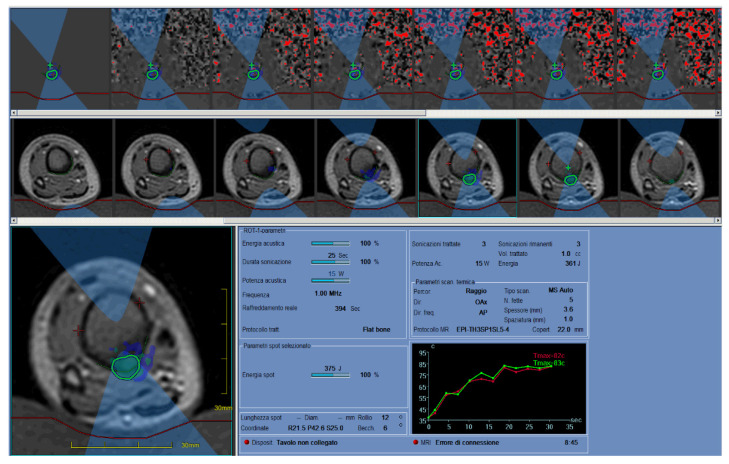
Tibial OB treated with MRgFUS was successful thanks to the presence of an adequate acoustic window and the superficial position of the lesion. Specific MR-sequences, acquired during the treatment, provided real-time thermometric maps of the region of interest to avoid thermal damage of the surrounding tissues.

## Data Availability

Data available in a publicly accessible repository.

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
