# Peer review of "Management of Osteoblastoma and Giant Osteoid Osteoma with Percutaneous Thermoablation Techniques"

_jcm, 2021, doi:10.3390/jcm10245717_

Round 1
Reviewer 1 Report
Title: OK.
Abstract:
OK
Keywords: OK
Informed Consent:
Ethical Committee (if relevant):
OK
Introduction:
OK
Radiofrequency ablation (RFA)
Pg. 4, ln 170. Reference 23 is referring cryoablation, and it is supposed to be about radiofrequency.
Pg.5, ln 172. When the authors state that the saline is used to insulate sensitive structures, it needs to be supported by a reference. I think in radiofrequency is more appropriate to use glucose or dextrose.
Figures.
OK
Statement:
This is a good review of the diagnosis and treatment of osteoblastoma. It is well written. Only two queries as stated above need to be answered.
Author Response
Point 1 Pg. 4, ln 170. Reference 23 is referring cryoablation, and it is supposed to be about radiofrequency.
Response 1. The reference is mainly about the thermoprotective measures: in that paper, in our opinion, the protective measures are very well described and because we looked for references about the treatment of Osteoblastoma with ablation techniques, we considered it suitable for citation, even if cryoablation was used as technique of ablation. However if you consider it unappropriate we can change it.
Point 2 Pg.5, ln 172. When the authors state that the saline is used to insulate sensitive structures, it needs to be supported by a reference. I think in radiofrequency is more appropriate to use glucose or dextrose.
Response 2. Yes, sorry. You are right, glucose solution have be used with RFA: we corrected the inaccuracy.
Reviewer 2 Report
The paper could be interesting for Scientifics and medical doctors. However, several aspects must be improved.
1.- It is not clear the main goal of this paper, why it is important to address about interventional Radiology’s Osteoblastoma management, if the osteoblastoma represents 1% of all primary bone tumors?
2.- The main goal of a review is divided in 3 aspects: 1) To investigate what is known about a topic, 2) what are the main investigated aspects, 3) What is still unknow. In the present review, none of these 3 aspects is clear.
3.- A systematic review is a process that help us to identify, evaluate and summarize the most relevant studies about a topic, in order to answer a research question. In the present review, the research question is not clear. Moreover, a more detailed and carefully analysis about the presented biographic must be done.
4.- The authors must describe the strategy follow up to collect the papers included in this study.
5.- The authors must describe Which data bases where consulted.
6.- What inclusion criterions were implemented to make the research?.
7.- The structure of the manuscript must be improved. In the current form, several aspects about the same topic are mixed in all the sections. The authors could divide the information in subsections, such as biological aspects, thermoablation physical aspects, thermoablation technical aspects, clinical aspects, RF applicators, MW applicators, US applicators, etc. It all depends about the main goal of the manuscript.
8.- Radiofrequency ablation is a bit more discussed. However, microwave ablation, Laser and US are barely analyzed.
9.- Discussion section must be added in order to address the relevance of this topic.
10.- The authors must address the advantage/disadvantage of each technique. Moreover, a comparison must be done.
11.- The authors could use tables in order to summarize and address the most relevant aspects.
Author Response
Dear reviewers, thank you for your revision. Here are the replies to your comments.
The paper could be interesting for Scientifics and medical doctors. However, several aspects must be improved.
1.- It is not clear the main goal of this paper, why it is important to address about interventional Radiology’s Osteoblastoma management, if the osteoblastoma represents 1% of all primary bone tumors?
Response 1. Osteoblastoma’s management is very important for MSK Interventional Ragiologists because, also if this pathology represents only 1% of all primary bone tumors, it always requires a therapy, because it is associated with an severe pain syndrome and worsening of the quality of life; currently, the gold standard treatment for this lesions is represented by thermoablation procedures.
2.- The main goal of a review is divided in 3 aspects: 1) To investigate what is known about a topic, 2) what are the main investigated aspects, 3) What is still unknow. In the present review, none of these 3 aspects is clear.
Response 2. Yes, these aspects reflect the most common scheme in reviews, but we have divided the paper in sections based on type of techniques to use.
3.- A systematic review is a process that help us to identify, evaluate and summarize the most relevant studies about a topic, in order to answer a research question. In the present review, the research question is not clear. Moreover, a more detailed and carefully analysis about the presented biographic must be done.
Response 3. This is a review about all the techniques used for the management of OB by Interventional Radiology.
4.- The authors must describe the strategy follow up to collect the papers included in this study.
Response 4. Yes, we done.
5.- The authors must describe Which data bases where consulted.
Response 5. Yes, we done.
6.- What inclusion criterions were implemented to make the research?.
Response 6. The inclusion criteria were: (1) prospective or retrospective cohort study for percutaneous treatment of OB or OO under CT guidance or MR guidance; (2) patient OB diagnosis by way of at least one CT or MR examination; (3) RFA, Cryoablation, MRgFUS and other thermoablation techniques as the object of the study; (4) English language; (5) follow-up ≥12 months.
7.- The structure of the manuscript must be improved. In the current form, several aspects about the same topic are mixed in all the sections. The authors could divide the information in subsections, such as biological aspects, thermoablation physical aspects, thermoablation technical aspects, clinical aspects, RF applicators, MW applicators, US applicators, etc. It all depends about the main goal of the manuscript.
Response 7. I can improve the manuscript by divinding the information in subsections, but this is not possible for all sections, because the available data in literature for some techniques are not adequate and sufficient.
8.- Radiofrequency ablation is a bit more discussed. However, microwave ablation, Laser and US are barely analyzed.
Response 8. Microwave ablation and laser ablation are barely analyzed because the experience with these techniques is very little and the literature is scarce and very fragmented, as also for MRgFUS.
9.- Discussion section must be added in order to address the relevance of this topic.
Response 9. The discussion is within each subsession (RFA, ….)
10.- The authors must address the advantage/disadvantage of each technique. Moreover, a comparison must be done.
Response 10. We have tried to describe advantages and disadvantages as possible with the evidence of the literature.
11.- The authors could use tables in order to summarize and address the most relevant aspects.
Response 11. We have tried, but the heterogeneous and fragmented data available for the other techniques apart the RFA make an eventual table of poor consultation.
Reviewer 3 Report
Dear authors, I appreciated your manuscript is well organized and concerning an important topic.
I suggest You some improvements:
- The title should be improved. I suggest You: e.g."The Role of Percutaneous Termoablation Techniques in the Management of Osteoblastoma and Giant Osteoid Osteoma"
- Since the radiological differentiation between giant osteoid osteoma and osteoblastoma is subtle (especially without follow-up control), I suggest You refer to both conditions.
- The most important concern is the possible malignant transformations. Even if it is a rare event, we cannot neglect this possibility. I suggest You to underline this concern and to add this as a limitation in the section dedicated to FUS treatment. Indeed, a biopsy is always suggested in these patients. In the reviewer's opinion this technique should be reserved for patients with less aggressive lesions appearance (e.g. not growing, osteoid osteoma like nidus, <2cm...). (suggested reference: Mesfin A, Boriani S, Gambarotti M, Bandiera S, Gasbarrini A. Can Osteoblastoma Evolve to Malignancy? A Challenge in the Decision-Making Process of a Benign Spine Tumor. World Neurosurg. 2020;136:150-156.) doi:10.1016/j.wneu.2019.11.148
- Figure 2. Are you sure this is an Osteoblastoma? Which criteria lead you to this diagnosis (diameter? in case specify it; growing?). If it is unclear please remove it. The appearance is of a regular osteoid osteoma.
- If You should provide a figure of cryoablation would be great (This is of course not mandatory being a literature review article).
Author Response
Dear reviewers, thank you for your revision. Here are the replies to your comments.
I suggest You some improvements:
1) The title should be improved. I suggest You: e.g."The Role of Percutaneous Termoablation Techniques in the Management of Osteoblastoma and Giant Osteoid Osteoma" Since the radiological differentiation between giant osteoid osteoma and osteoblastoma is subtle (especially without follow-up control), I suggest You refer to both conditions.
Response 1. Ok, we done
2) The most important concern is the possible malignant transformations. Even if it is a rare event, we cannot neglect this possibility. I suggest You to underline this concern and to add this as a limitation in the section dedicated to FUS treatment. Indeed, a biopsy is always suggested in these patients. In the reviewer's opinion this technique should be reserved for patients with less aggressive lesions appearance (e.g. not growing, osteoid osteoma like nidus, <2cm...). (suggested reference: Mesfin A, Boriani S, Gambarotti M, Bandiera S, Gasbarrini A. Can Osteoblastoma Evolve to Malignancy? A Challenge in the Decision-Making Process of a Benign Spine Tumor. World Neurosurg. 2020;136:150-156.) doi:10.1016/j.wneu.2019.11.148
Response 2. Yes, you are right, I’ll add this limitations in the section dedicated to FUS treatment.
3) Figure 2. Are you sure this is an Osteoblastoma? Which criteria lead you to this diagnosis (diameter? in case specify it; growing?). If it is unclear please remove it. The appearance is of a regular osteoid osteoma.
Response 3. We reported this case because the histological response was osteoblastoma. However we can remove it if you prefer.
4) If You should provide a figure of cryoablation would be great (This is of course not mandatory being a literature review article)
Response 4. I’m sorry, but we don’t have a figure of OB cryoablation, because we don’t have a personal experience of this technique in the treatment of OB, in our Institute.
Round 2
Reviewer 2 Report
Although the topic is interesting for the Scientifics in the field; it was not properly discussed. The authors include just some of my previous recommendations; however, this information was not enough to show the relevance of this manuscript. Therefore, my recommendation to the authors is to rethink about the main goal of this work and then report the most relevant information that address the relevance and the necessity of this work.
Author Response
Dear revisor, thank you for your suggestions and comments, we have tried to do every effort to improve the mauscript as suggested bu you.
Best Regards
Reviewer 3 Report
I'm satisfied with the revisions performed,
Thank You.
Author Response
Dear revisor, thank you for your suggestions and comments.
Best Regards